

# Efficacy of a new dietary supplement in dogs with advanced chronic kidney disease

Elisa Martello[1], Francesca Perondi[2], Maria Teresa Capucchio[3], Ilaria Biasato[4], Elena Biasibetti[3], Tiziana Cocca[5], Natascia Bruni[6] and Ilaria Lippi[2]

[1] Division of Epidemiology and Public Health, School of Medicine, University of Nottingham, Nottingham, United Kingdom
[2] Department of Veterinary Science, University of Pisa, San Piero a Grado (PI), Italy
[3] Department of Veterinary Sciences, University of Turin, Grugliasco (TO), Italy
[4] Department of Agricultural, Forest and Food Sciences, University of Turin, Grugliasco (TO), Italy
[5] Clinica Veterinaria Napolivet, Napoli, Italy
[6] Candioli Pharma S.r.l., Beinasco (TO), Italy

## ABSTRACT

Chronic kidney disease (CKD) is a common disease in elderly dogs. The present study aims to evaluate the efficacy of a dietary supplement containing calcium carbonate, calcium-lactate gluconate, chitosan and sodium bicarbonate in dogs with IRIS stage 3 of CKD. Twenty dogs were enrolled in the study, ten were administered the new dietary supplementation for 180 days (T group) while the others were used as control group (C group). Haematologic, biochemical and urinalysis were performed every 30 days. A significant reduction in the T group compared to the C group in serum phosphorus level and increase in serum bicarbonate and ionized calcium values were recorded. The urine protein-to-creatinine ratio (UPC) was significantly lower in the T group at the end of the study compared to the C group. The tested supplement could be considered as a supportive treatment for dogs with advanced CKD.

## INTRODUCTION

Chronic kidney disease (CKD) is generally defined as any structural and/or functional abnormality, which can affect one or both kidneys and has been present in patients for at least three months. Chronic kidney disease is a very common disease in canine population, with a high prevalence especially in older patients. Generally, the progression of the disease is slow, with a survival time from months to a year or two (*Bartges, 2012*; *Davies, 2016*; *Polzin, 2011*; *Smets et al., 2010*). The management of CKD is focused on slowing down the progression of the disease by controlling the major risk factors and clinical signs. In veterinary medicine there are several parameters to be monitored overtime in patients with CKD in order to modify the therapy when needed. The major factors to monitor are: proteinuria, hypertension and hyperphosphatemia.

Corresponding author
Elisa Martello,
msaem7@exmail.nottingham.ac.uk,
martello.elisa@gmail.com

Renal diets specifically designed for CKD aims to slow down progression of the disease and extend survival, to improve clinical consequences of uremia and minimize electrolyte and acid–base unbalance, and maintain adequate nutritional intake (*Biasibetti et al., 2018*; *Jacob et al., 2002*; *Polzin, 2013*). The formulation of canine renal diets has changed and improved over the years, the administration of this food is considered the first therapeutic approach for improving survival and life quality of canine patients with CKD, using a four-stage scale of disease progression according to the International Renal Interest Society (IRIS) guidelines (*IRIS, 2019*; *Davies, 2016*). Nutritional support is also essential to control the protein and phosphate intake to avoid the risk of malnutrition and dehydration and to reduce mortality (*Rudinsky et al., 2018*). When the diet alone is not sufficient to control phosphate and/or metabolic acidosis or to slow down the progression of CKD, the use of dietary supplements (i.e., phosphate binders and alkalizing agents) is a useful strategy adopted by veterinarians (*Lippi et al., 2017*; *Polzin, 2011*; *Zatelli et al., 2017*).

The clinicians have to choose the most appropriate therapeutic approach to maintain serum phosphorus concentration within the range reported by the IRIS, according to each CKD stage. Currently, there is a wide choice of commercially available phosphate binders which, through mechanisms of chelation, are able to prevent the absorption of phosphorus by the intestinal mucosa, through the formation of inert compounds directly eliminated with the faeces. Metabolic acidosis is another promoter for the progression of CKD and for impairing protein nutrition (*De Brito-Ashurst et al., 2009*; *Polzin, 2013*). With CKD, there is increased retention of metabolic acids, increased production of ammonia, and decreased production of bicarbonate. Metabolic acidosis occurs in less than 10% of cats with stage 2 or 3 CKD and in 50% with uremia (*Bartges, 2012*; *Polzin, 2013*), but to our knowledge no data is available for prevalence in dogs with CKD. Metabolic acidosis is associated with hyporexia/anorexia, hypokalemia, and muscle weakness (*Bartges, 2012*). In human medicine, therapy with bicarbonate has been reported to slow down the progression of CKD and to improve the nutritional and health status (*Raphael, 2019*). Metabolic alkalinizing agents are commonly used to increase the blood pH given the ability to bind H+ ions. Alkalization therapy is indicated for dogs with IRIS CKD Stages 1–4 when blood pH and bicarbonate concentration drop below the normal range. When diet alone is insufficient, administration of alkalinizing agents such as potassium citrate and sodium bicarbonate is indicated (*IRIS , 2019*; *Polzin, 2013*; *Zatelli et al., 2017*; *Zatelli et al., 2012*).

In dogs with CKD, commercial dietary supplements with different compounds such as chitosan calcium carbonate, and potassium citrate show their beneficial effect to control hyperphosphatemia (*Zatelli et al., 2012*). A recent study on cats testing an oral supplement with calcium carbonate, calcium-lactate gluconate, chitosan and also sodium bicarbonate confirms the reduced serum phosphorus and the increase in serum bicarbonate with improvement of clinical conditions (*Biasibetti et al., 2018*).

The present study aims to evaluate the ability to bind phosphate and correct metabolic acidosis, together with the ease of administration of a new commercial dietary supplement containing calcium carbonate, calcium-lactate gluconate, chitosan and sodium bicarbonate in dogs with CKD.

## MATERIALS & METHODS

A prospective, randomised, controlled study on dogs affected by CKD was performed in 2015. The Veterinary clinic Napolivet (Naples, Italy) database was screened in order to include dogs with advanced CKD stages 3 and/or 4 according to the International Renal Interest Society (IRIS). Diagnosis of CKD was performed by persistently azotemia by previous laboratory findings and/or estimated from the medical history ($> 3$ months of polyuria/polydipsia), physical examination findings consistent with chronic disease (loss of weight and lean body mass, small kidney size), or evidence of chronic structural abnormalities identified by ultrasound imaging (small kidneys, renal infarcts, renal fibrosis). Dogs included were classified following guidelines developed by IRIS, based on serum creatinine concentration, the magnitude of proteinuria as measured by the urine protein, creatinine ratio (UPC) and blood pressure (BP) (*IRIS, 2013*).

Dogs suspected or affected by other concomitant diseases (acute kidney injury, pre-renal or post-renal azotemia, genitourinary tract inflammation or infection, urinary tract obstruction, heart disease, chronic heart failure, neoplasia, hypothyroidism, diabetes) assessed by clinical, instrumental and laboratory evaluations, were excluded from the study.

The dogs' owners were informed about the purpose and the design of the study and signed a written informed consent. All procedures, treatments and animal care were in compliance with the guidelines of the Italian Minister of Health for the care and use of animals (D.L. 4 March 2014 n. 26 and D.L. 27 January 1992 n.116) and UE (Directive 86/609/CEE) and the use of supplements was regulated by the Regulation (EC) No 767/2009.

After the enrollment in the study (T0), dogs were randomly assigned to two groups: the control group (C group, $n = 10$) and the treated group (T group, $n = 10$). History, physical examination including body weight (BW) and body condition score (BCS) (1 to 5 scoring system) were recorded. Complete blood count (CBC), serum biochemical profile, venous blood gas analysis were evaluated. Blood pressure (BP) was taken by an sphygmomanometers (CONTEC), during the examination, measurements were taken five times and the mean value was recorded. Urinalysis procedures were already described in Biasibetti and colleagues (*Biasibetti et al., 2018*). A complete clinical examination, blood and urinary tests were performed at the beginning of the study (T0), then at day 30 (T30), 60 (T60), 90 (T90), 120 (T120), 150 (T150) and 180 (T180). All the animals were fed with the same commercial renal diet (Royal Canin Renal Canine) for the whole duration of the trial and from at least eight weeks before the inclusion in the study. The amount offered to each dog was based on the estimated metabolic requirements (calorie intake) according to the FEDIAF Nutritional Guidelines (Fédération européenne de l'industrie des aliments pour animaux familiers (FEDIAF), 2014). In the T group a new supplement produced by the Candioli Pharma S.r.l. ("Renal P") containing calcium carbonate, calcium-lactate gluconate, chitosan and sodium bicarbonate (Table 1) was added to the commercial renal diet. The daily supplement dosage was set as 0.2 g/kg body weight, and it was divided in two

**Table 1 Composition of the feed supplement used during the study.**

| COMPOSITION | g/100 g |
| --- | --- |
| Calcium Lactate Gluconate | 16.00 |
| Calcium carbonate | 26.00 |
| Sodium bicarbonate | 5.00 |
| Chitosan | 8.00 |
| Silica | 0.50 |
| Maltodextrin | 44.30 |
| Sodium pyrophosphate | 0.08 |
| Yeasts [Brewer'yeasts] | 0.05 |
| Lupin meal | 0.02 |
| Salts | 0.02 |
| Sunflower oil | 0.01 |
| **TOTAL** | **100** |

daily administrations mixed with the two meals, so as to favor the reduction of intestinal phosphorus absorption. The supplement was given for a total of 180 days. nata.

Dogs with clinical signs (vomiting, poor appetite) and/or proteinuria, hypertension, were started on appropriate treatment weeks to months prior to T0. Owners were asked to monitor daily eposodes of vomiting, diarrhea, anorexia during the entire duration of the study.

Dogs with persistent proteinuria were treated with benazepril (Fortekor; Novartis Animal Health, Varese, Italy), 0.25 to 0.5 mg/kg body weight (PO) once to twice daily. Dogs with a history of hypertension (BP > 160 mmHg) and/or with the risk of extra-renal target organ damage (TOD) were maintained on a combination of benazepril and amlodipine (Amodip; CEVA Salute Animale), 0.25 to 0.5 mg/kg BW (PO) once daily as recommended by IRIS staging system (*IRIS, 2013*).

## Blood examination

During each examination, blood examination and urinalysis were performed.

The laboratory performed CBC (Animal blood counter, Scil -Vet abcTM). Serum biochemical analysis (Automatic Analyzer-Echo, Edif) recorded the values of: blood urea nitrogen (BUN), creatinine (CREA), phosphorus (P), total protein (TP), albumin (ALB), albumin/globulin ratio (A/G), glucose (GLU), alanine aminotransferase (ALT), aspartate aminotransferase (AST), alkaline phosphatase (ALP), bilirubin (BIL), and cholesterol (CHOL). Venous blood gas analysis was also performed in all cases with an analytical device (Abaxis VetScan i-Stat1) in order to assess bicarbonate (HCO3) and ionized calcium (iCA).

## Urinalysis

Urine samples were obtained by cystocentesis with a five mL syringe. They were put in a sterile collection tube and analyzed. Urine sediment was obtained by centrifugation (2 min at 1,500 × g) followed by removal of supernatant which was stored at +4 °C. For the urine sediment examination and specific gravity test an in-house refractometer was used.

**Table 2  Body weight (BW) and Body Condition Score (BCS) measured during the study in control (C) and treated (T) groups.** Data are expressed as mean and standard error of the mean (SEM); T, time in day.

| Parameter | T0 | T30 | T60 | T90 | T120 | T150 | T180 |
|---|---|---|---|---|---|---|---|
| | | | | Group C | | | |
| BW kg | 21.44 ± 3.50 | 21.20 ± 3.56 | 20.82 ± 3.58 | 20.48 ± 3.56 | 23.07 ± 3.68 | 22.39 ± 3.49 | 21.69 ± 3.41 |
| BCS | 2.89 ± 0.35 | 2.78 ± 0.36 | 2.67 ± 0.37 | 2.56 ± 0.38 | 2.89 ± 0.31 | 2.89 ± 0.31 | 2.56 ± 0.34 |
| | | | | Group T | | | |
| BW kg | 17.76 ± 3.28 | 17.75 ± 3.38 | 17.96 ± 3.43 | 17.72 ± 3.39 | 17.82 ± 3.41 | 17.85 ± 3.41 | 17.91 ± 3.41 |
| BCS | 2.6 ± 0.16 | 2.70 ± 0.15 | 2.70 ± 0.15 | 2.70 ± 0.15 | 2.70 ± 0.15 | 2.80 ± 0.13 | 2.80 ± 0.13 |

To calculate the UPC, protein and creatinine concentrations (mg/dL) were measured with the pyrogallol red method and Jaffe' method, respectively, within 12 h from the collection. Dipstick analysis was used for the urine test strip.

Dogs were classified as non-proteinuric, borderline proteinuric, or proteinuric according to the IRIS substaging system (UPC < 0.2 = non-proteinuric, UPC 0.2 to 0.5 = borderline proteinuric, and UPC > 0.5 = proteinuric).

## Statistical analysis

GraphPad Prism® software was used to perform statistical analysis. Shapiro–Wilk test established the normality or non-normality distribution of data. Repeated measures ANOVA (post-hoc test: Bonferroni's Multiple Comparison Test) and Friedman (post-hoc test: Dunn's Multiple Comparison test) tests were used to perform the intra-group comparisons among the experimental times, while the inter-group comparisons between groups were performed by Student t and Mann–Whitney U tests (with $p$-values corrected to account for multiple comparisons). Data are expressed as mean and SEM. Significance was set as $p < 0.05$ (intra-group comparisons) or $p < 0.008$ (inter-group comparisons).

## RESULTS

A total of 20 dogs with CKD IRIS stage 3 were recruited and all dogs ($n = 10$ T group; $n = 10$ C group) completed the study. The median age of all dogs was 10 years. Dogs' ages ranged from 5 to 15 years (mean $10.7 ± 2.9$ SD) in the T group and from 8 to 14 years (mean $10.1 ± 1.8$ SD) in the C group. In the T group, four dogs were intact males and six were females (50% sterilized), while in the C group six dogs were intact males and four were females (50% sterilized). Eight dogs were mixed breed, two were German Shepherd, then one was Maltese, one Belgian Shepherd, one Yorkshire terrier, one Levriero, one Fox Terrier, one Jack Russel Terrier, one Schnauzer, one Beagle, one Dobermann and one Basset hound. At the time of inclusion as well as at the end of the study, there were no statistical differences between ($p > 0.008$) or ($p > 0.05$) within groups with regards to BW and BCS (Table 2).

Hematological values (Table 3) were recorded during the entire study. All the hematological parameters were also similar between the two groups for all the considered experimental times ($p > 0.008$). Furthermore, no significant differences were recorded

**Table 3  Hematochemical parameters: haematocrit (HCT), haemoglobin (HG), red blood cells (RBC), white blood cells (WBC), neutrophil (N), eosinophil (EO), lymphocytes (LYM) measured during the study in control and treated groups.** Data are expressed as mean and standard error of the mean (SEM); T, time in days.

| Parameter | Laboratory standard reference range | T0 | T30 | T60 | T90 | T120 | T150 | T180 |
|---|---|---|---|---|---|---|---|---|
| | | | | Parameters group C | | | | |
| HCT % | 37–55 | 30.76 ± 2.43 | 29.86 ± 2.07 | 29.71 ± 2.48 | 28.40 ± 2.20 | 29.87 ± 2.37 | 28.42 ± 2.30 | 27.59 ± 2.34 |
| HG g/dL | 12–18 | 12.34 ± 0.67 | 12.83 ± 0.57 | 12.66 ± 0.74 | 11.79 ± 0.71 | 12.59 ± 0.88 | 12.09 ± 0.82 | 11.76 ± 0.80 |
| RBC $10^6$ $mm^3$ | 5.5–8.5 | 5.80 ± 0.33 | 5.96 ± 0.28 | 5.93 ± 0.32 | 5.71 ± 0.27 | 5.92 ± 0.31 | 5.68 ± 0.31 | 5.49 ± 0.29 |
| WBC $10^3$ $mm^3$ | 6-17 | 11.27 ± 1.09 | 11.29 ± 1.12 | 11.17 ± 1.63 | 10.24 ± 1.25 | 10.74 ± 0.94 | 10.61 ± 1.25 | 10.02 ± 0.68 |
| N $10^3$ $mm^3$ | 3–11.5 | 7.74 ± 0.74 | 7.77 ± 0.78 | 7.71 ± 1.07 | 6.94 ± 0.95 | 7.17 ± 0.61 | 7.26 ± 1.02 | 6.69 ± 0.48 |
| EO $10^3$ $mm^3$ | 0.1-1.3 | 0.97 ± 0.07 | 1.12 ± 0.12 | 1.07 ± 0.17 | 0.98 ± 0.10 | 1.04 ± 0.16 | 0.98 ± 0.08 | 1.06 ± 0.08 |
| LYM $10^3$ $mm^3$ | 1–4.8 | 1.44 ± 0.18 | 1.29 ± 0.10 | 1.34 ± 0.21 | 1.41 ± 0.20 | 1.37 ± 0.24 | 1.28 ± 0.12 | 1.32 ± 0.15 |
| | | | | Parameters group T | | | | |
| HCT % | 37–55 | 33.14 ± 3.61 | 32.81 ± 2.82 | 32.76 ± 2.62 | 31.25 ± 2.71 | 31.25 ± 2.57 | 32.38 ± 2.56 | 33.22 ± 2.50 |
| HG g/dL | 12–18 | 12.27 ± 1.24 | 12.23 ± 0.93 | 12.01 ± 0.85 | 11.46 ± 0.83 | 11.48 ± 0.78 | 14.61 ± 2.72 | 15.10 ± 2.77 |
| RBC $10^6$ $mm^3$ | 5.5–8.5 | 5.72 ± 0.48 | 5.67 ± 0.28 | 5.65 ± 0.30 | 5.49 ± 0.33 | 5.51 ± 0.31 | 5.80 ± 0.27 | 6.04 ± 0.26 |
| WBC $10^3$ $mm^3$ | 6–17 | 11.04 ± 1.57 | 10.17 ± 0.87 | 8.81 ± 0.65 | 8.44 ± 0.75 | 9.17 ± 0.63 | 9.55 ± 0.57 | 9.61 ± 0.77 |
| N $10^3$ $mm^3$ | 3-11.5 | 7.83 ± 1.15 | 6.88 ± 0.85 | 5.81 ± 0.66 | 5.86 ± 0.66 | 5.83 ± 0.56 | 6.54 ± 0.61 | 6.11 ± 0.45 |
| EO $10^3$ $mm^3$ | 0.1–1.3 | 0.96 ± 0.16 | 1.00 ± 0.11 | 0.95 ± 0.09 | 1.04 ± 0.07 | 1.34 ± 0.11 | 1.34 ± 0.11 | 1.32 ± 0.08 |
| LYM $10^3$ $mm^3$ | 1-4.8 | 1.09 ± 0.20 | 1.35 ± 0.26 | 1.28 ± 0.25 | 1.23 ± 0.16 | 1.39 ± 0.22 | 1.35 ± 0.12 | 1.23 ± 0.13 |

**Table 4  Blood pressure measured during the study. Data are expressed as mean and standard error of the mean (SEM).**

| | T0 | | T180 | |
|---|---|---|---|---|
| | Diastolic (mm Hg) | Systolic (mm Hg) | Diastolic (mm Hg) | Systolic (mm Hg) |
| Group C | 86.67 ± 3.27 | 151.67 ± 4.66 | 85.00 ± 1.69 | 153.75 ± 3.36 |
| Group T | 85.00 ± 2.69 | 136.00 ± 5.82 | 84.00 ± 1.63 | 150.50 ± 2.41 |

for blood pressure between or within groups at the beginning and at the end of the study ($p > 0.05$) (Table 4).

All the biochemical parameters measured during the study are reported in Table 5. Most of the parameters remained stable during the trial in both groups. However, an increase in blood urea nitrogen (BUN) ($p < 0.05$) and creatinine (CREA)($p < 0.05$) was reported in the C group, while BUN remained stable and CREA progressively decreased in the T group during the study period ($p < 0.05$). Even so, all these parameters were similar between the two groups for all the considered experimental times ($p > 0.008$).

No difference in the serum phosphorus level between the two groups was reported at day 0 ($p > 0.008$). This value significantly decreased at each time point from day 30 to 180 ($p < 0.008$) in the T group compared to the C group (Fig. 1). In addition, in the C group all dogs, at each time point, showed levels of serum phosphorus above the normal range (2.7–5 mg/dl), while in seven dogs in the T group the same level came back to normal

Martello et al. (2020), *PeerJ*, DOI 10.7717/peerj.9663

**Table 5 Biochemical parameters measured during the study: blood urea nitrogen (BUN), creatinine (CREA), phosphorus (P), total 3 protein (TP), albumin (ALB), albumin/globulin (A/G), glucose (GLU), alanine aminotransferase (ALT), aspartate aminotransferase 4 (AST), alkaline phosphatase (ALP), bilirubin (BIL), cholesterol (CHOL).** Data are expressed as mean and standard error of the mean 5 (SEM); T: time in days. Means with different superscript letters (a, b, c) indicate significant differences ($P < 0.05$) among experimental 6 times within each group per each investigated parameter.

| Parameter | Laboratory standard reference range | T0 | T30 | T60 | T90 | T120 | T150 | T180 |
|---|---|---|---|---|---|---|---|---|
| | | | | Parameters group C | | | | |
| BUN mg/dL | 15–45 | $120 \pm 10.87^a$ | $132.00 \pm 14.05^a$ | $132.56 \pm 12.54^a$ | $139.00 \pm 10.42^{ab}$ | $132.00 \pm 9.91^{ab}$ | $136.00 \pm 9.75^{ab}$ | $145.44 \pm 13.29^b$ |
| CREA mg/dL | 0.5–1.8 | $3.14 \pm 0.23$ | $3.16 \pm 0.21$ | $3.17 \pm 0.15$ | $3.33 \pm 0.13$ | $3.29 \pm 0.19$ | $3.36 \pm 0.14$ | $3.46 \pm 0.18$ |
| P mg/dL | 2.7–5 | $7.81 \pm 0.24$ | $8.09 \pm 0.37$ | $8.14 \pm 0.22$ | $8.32 \pm 0.21$ | $8.00 \pm 0.25$ | $8.16 \pm 0.23$ | $8.23 \pm 0.22$ |
| TP mg/dL | 6–7.5 | $7.28 \pm 0.43^a$ | $7.94 \pm 0.64^a$ | $7.71 \pm 0.61^a$ | $7.40 \pm 0.58^{ab}$ | $7.91 \pm 0.68^{ab}$ | $7.76 \pm 0.60^{ab}$ | $7.61 \pm 0.61^b$ |
| ALB mg/dL | 2.5–4.2 | $3.16 \pm 0.13$ | $3.12 \pm 0.12$ | $3.09 \pm 0.08$ | $3.00 \pm 0.07$ | $3.21 \pm 0.09$ | $3.13 \pm 0.08$ | $3.03 \pm 0.06$ |
| A/G mg/dL | 0.5–1.3 | $0.91 \pm 0.07$ | $0.82 \pm 0.10$ | $0.83 \pm 0.08$ | $0.84 \pm 0.08$ | $0.79 \pm 0.08$ | $0.84 \pm 0.07$ | $0.82 \pm 0.07$ |
| GLU mg/Dl | 50–100 | $86.89 \pm 4.00$ | $90.78 \pm 4.45$ | $87.89 \pm 3.15$ | $89.44 \pm 4.49$ | $88.11 \pm 6.32$ | $85.89 \pm 3.55$ | $86.00 \pm 4.63$ |
| ALT UI/L | 7–40 | $60.89 \pm 10.26$ | $71.89 \pm 19.63$ | $68.33 \pm 16.24$ | $63.89 \pm 15.05$ | $72.00 \pm 12.70$ | $69.67 \pm 11.16$ | $67.11 \pm 10.95$ |
| AST UI/L | 7–40 | $54.78 \pm 9.21$ | $54.00 \pm 8.38$ | $52.44 \pm 8.53$ | $49.56 \pm 7.07$ | $52.67 \pm 6.94$ | $52.11 \pm 5.79$ | $53.56 \pm 6.44$ |
| ALP UI/L | 5–110 | $191.78 \pm 22.39$ | $203.89 \pm 26.17$ | $211.33 \pm 30.15$ | $216.33 \pm 32.61$ | $212.67 \pm 25.25$ | $201.67 \pm 23.71$ | $208.33 \pm 20.66$ |
| BIL mg/dL | 0–0.7 | $0.15 \pm 0.02$ | $0.18 \pm 0.03$ | $0.18 \pm 0.03$ | $0.18 \pm 0.04$ | $0.17 \pm 0.03$ | $0.18 \pm 0.03$ | $0.18 \pm 0.03$ |
| CHOL mg/dL | 140–240 | $384.67 \pm 23.84$ | $406.33 \pm 24.44$ | $407.89 \pm 23.26$ | $413.78 \pm 25.95$ | $386.56 \pm 24.14$ | $406.22 \pm 30.32$ | $399.33 \pm 27.66$ |
| | | | | Parameters group T | | | | |
| BUN mg/dL | 15–45 | $124.45 \pm 12.94$ | $110.23 \pm 10.87$ | $107.53 \pm 9.11$ | $115.41 \pm 10.74$ | $123.74 \pm 12.14$ | $119.24 \pm 10.35$ | $124.51 \pm 12.12$ |
| CREA mg/dL | 0.5–1.8 | $3.52 \pm 0.29^a$ | $3.05 \pm 0.19^b$ | $3.03 \pm 0.16^b$ | $3.15 \pm 0.17^{ab}$ | $3.14 \pm 0.15^{ab}$ | $3.12 \pm 0.11^{ab}$ | $3.19 \pm 0.09^{ab}$ |
| P mg/dL | 2.7–5 | $8.76 \pm 0.39^a$ | $4.73 \pm 0.35^b$ | $4.16 \pm 0.23^{bc}$ | $3.86 \pm 0.18^{bc}$ | $3.70 \pm 0.14^c$ | $3.60 \pm 0.13^c$ | $3.59 \pm 0.12^c$ |
| TP mg/dL | 6–7.5 | $7.30 \pm 0.56$ | $7.02 \pm 0.37$ | $6.79 \pm 0.33$ | $6.70 \pm 0.34$ | $6.60 \pm 0.31$ | $6.72 \pm 0.29$ | $6.74 \pm 0.25$ |
| ALB mg/dL | 2.5–4.2 | $3.12 \pm 0.16$ | $3.24 \pm 0.10$ | $3.14 \pm 0.10$ | $3.04 \pm 0.10$ | $2.96 \pm 0.13$ | $3.02 \pm 0.11$ | $3.03 \pm 0.09$ |
| A/G mg/dL | 0.5–1.3 | $0.76 \pm 0.08$ | $0.77 \pm 0.05$ | $0.84 \pm 0.05$ | $0.90 \pm 0.04$ | $0.90 \pm 0.05$ | $0.88 \pm 0.05$ | $0.86 \pm 0.05$ |
| GLU mg/Dl | 50–100 | $87.27 \pm 4.97$ | $91.66 \pm 2.95$ | $88.46 \pm 2.82$ | $86.40 \pm 2.53$ | $92.50 \pm 2.88$ | $84.90 \pm 2.36$ | $83.1 \pm 3.87$ |
| ALT UI/L | 7–40 | $63.01 \pm 6.95^a$ | $53.63 \pm 3.38^{ab}$ | $50.61 \pm 3.27^b$ | $46.35 \pm 4.00^b$ | $46.71 \pm 3.43^b$ | $44.60 \pm 3.53^b$ | $43.90 \pm 2.44^b$ |
| AST UI/L | 7–40 | $48.76 \pm 5.09$ | $46.00 \pm 3.90$ | $45.30 \pm 4.06$ | $43.21 \pm 3.19$ | $42.64 \pm 2.52$ | $46.00 \pm 6.46$ | $40.80 \pm 2.49$ |
| ALP UI/L | 5–110 | $137.93 \pm 21.08^a$ | $152.81 \pm 22.63^{ab}$ | $156.39 \pm 21.69^{ab}$ | $181.50 \pm 20.93^b$ | $170.43 \pm 18.89^{ab}$ | $178.40 \pm 18.30^b$ | $188.20 \pm 17.68^b$ |
| BIL mg/dL | 0–0.7 | $0.29 \pm 0.10$ | $0.26 \pm 0.08$ | $0.23 \pm 0.06$ | $0.21 \pm 0.05$ | $0.22 \pm 0.06$ | $0.21 \pm 0.05$ | $0.20 \pm 0.04$ |
| CHOL mg/dL | 140–240 | $318.58 \pm 24.39$ | $311.45 \pm 23.24$ | $332.56 \pm 24.82$ | $332.80 \pm 28.04$ | $335.70 \pm 25.57$ | $334.00 \pm 23.35$ | $332.30 \pm 24.19$ |

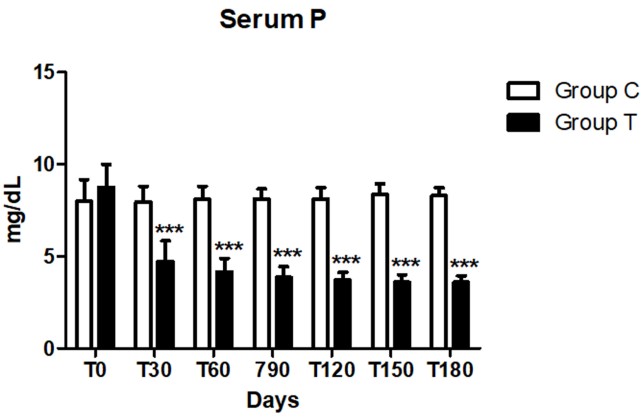

**Figure 1** Serum phosphorus (P) concentration at days 0, 30, 60, 90, 120, 150 and 180 (*** $p < 0.001$).

**Table 6** Venous haemogas analysis parameters (serum bicarbonate ($HCO_3$) and ionized calcium (iCa)) measured during the study at regular intervals. Data are expressed as mean and standard error of the mean (SEM); T: time in days. In the T group, means with different superscript letters (a, b, c) indicate significant differences ($p < 0.05$) among the experimental times.

| Parameter | Laboratory standard reference range | T0 | T30 | T60 | T90 | T120 | T150 | T180 |
|---|---|---|---|---|---|---|---|---|
| | | | | Parameters group C | | | | |
| HCO3- mmol/L | 18–24 | 16.86 ± 1.38 | 16.93 ± 1.30 | 16.78 ± 1.17 | 16.56 ± 1.02 | 16.64 ± 0.99 | 16.58 ± 0.89 | 16.36 ± 0.81 |
| iCa mmol/L | 1.29–1.41 | 1.27 ± 0.06 | 1.27 ± 0.06 | 1.27 ± 0.06 | 1.27 ± 0.06 | 1.26 ± 0.06 | 1.25 ± 0.05 | 1.25 ± 0.06 |
| | | | | Parameters group T | | | | |
| HCO3- mmol/L | 18–24 | 16.25 ± 0.50[a] | 16.69 ± 0.49[b] | 17.01 ± 0.45[c] | 17.02 ± 0.42[c] | 17.06 ± 0.35[c] | 17.10 ± 0.30[c] | 17.15 ± 0.30[c] |
| iCa mmol/L | 1.29–1.41 | 1.24 ± 0.04[a] | 1.30 ± 0.05[ab] | 1.31 ± 0.05[ab] | 1.32 ± 0.05[b] | 1.34 ± 0.04[b] | 1.33 ± 0.03[b] | 1.34 ± 0.03[b] |

within 30 days after the beginning of the treatment, in the other three at T60 ($n = 1$) and T90 ($n = 2$) and it remained so during the course of the treatment. Considering the mean values, a significant progressive reduction overtime was reported in the T group ($p < 0.05$) (Fig. 1, Table 5).

No difference in the serum iCa level between the two groups was observed at day 0 ($p > 0.008$). A significant increase of this parameter was observed from day 120 ($p < 0.008$) in the T group compared with the C group, but it remained within the normal range (1.29–1.41 mmol/l) *Bachmann et al. (2018)* while the mean iCa level in the C group remained stable during the study and below the reference range (Table 6, Fig. 2).

No difference in the serum $HCO_3$ level between the two groups was reported at day 0 ($p > 0.008$). Serum $HCO_3$ was lower than the normal range (18–24 mmol/l) in both groups at T0 (Table 6), but limited to the T group we observed a significant increase in values during the trial from day 30 on ($p < 0.005$). As a result, at day 180, $HCO_3$ was significantly

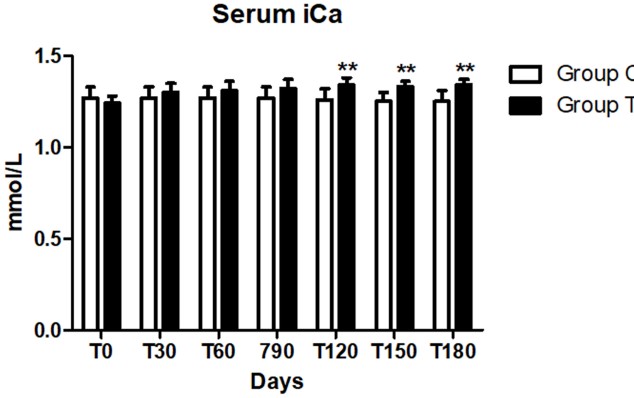

**Figure 2** Serum ionized calcium (iCa) concentration at days 0, 30, 60, 90, 120, 150 and 180 (**$p < 0.01$).

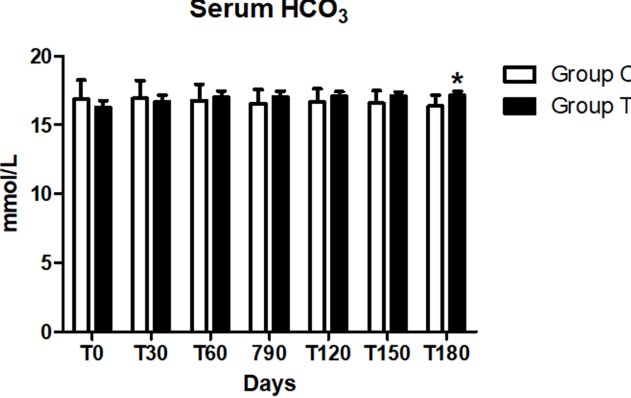

**Figure 3** Serum bicarbonate ($HCO_3$) concentration at days 0, 30, 60, 90, 120, 150 and 180 (*$p < 0.05$).

greater ($p < 0.008$) in the T group than in the C group (Table 6A, Fig. 3), but mean values were still below the reference range.

At the baseline, all dogs ($n = 10$) of the C group and eight dogs in the T group were proteinuric (UPC > 0.5) , two were borderline proteinuric (UPC 0.2–0.5) (Table 7). However, no differences in the UPC between the two groups were observed at day 0 ($p > 0.008$). On the contrary, urine protein, creatinine ratio at days 150 and 180 was lower ($p < 0.008$) in the T group compared to the C group (Fig. 4). No differences in UPC, were found within the same group during the treatment period ($p < 0.05$).

All the dogs remained in IRIS stage 3 for the duration of the study. All the owners in the T group, reported the ease of administration of the supplement which was entirely consumed as specifically indicated in the study protocol.

All the selected animals completed the study and all the owners reported a good palatability of the tested feed supplementation product. No adverse effects (vomiting,

**Table 6B** Venous haemogas analysis parameters (serum bicarbonate, $HCO_3$) measured at the beginning (T0) and the end (T180) of the study in the single patients from the T group.

| Patients | T0 | T180 |
|---|---|---|
| 1 | 16,8 | 17,3 |
| 2 | 17,0 | 17,5 |
| 3 | 16,4 | 17,1 |
| 4 | 16,4 | 17,2 |
| 5 | 16,8 | 17,7 |
| 6 | 15,9 | 17,1 |
| 7 | 15,8 | 17,0 |
| 8 | 15,8 | 17,1 |
| 9 | 16,0 | 16,7 |
| 10 | 15,6 | 16,8 |

diarrhea, anorexia) monitored daily by the owner were observed during the whole study period.

## DISCUSSION

The main aim of the present study was to determine the efficacy of a new dietary supplement administered for a period of 180 days in dogs in IRIS stage 3. In the present study, the time of observation was longer (180 vs 28 days) than another study where a similar supplement contained chitosan, enteric phosphate binders, and alkalinizing agents was used (*Zatelli et al., 2012*), it is a great advantage in terms of monitoring the long-term impact and safety of the product. The supplement administration helped reducing the mortality rate due to uremic crises in dogs with CKD, but it failed to show significantly improvement for mean serum concentrations of creatinine, BUN, phosphate and $HCO_3$ (*Zatelli et al., 2012*). In addition, the values were recorded at 4–8 weeks following the supplement administration (*Zatelli et al., 2012*). A more recent study, showed that another type of dietary supplement reduced significantly serum phosphorus and increased $HCO_3$ values in cats with CKD, improving their clinical conditions, without any adverse reaction (*Biasibetti et al., 2018*). In our study, the dogs treated with dietary supplementation showed a significant reduction in serum phosphorus level and an increase in $HCO_3$ compared to the control group (Fig. 1). In particular, mean serum phosphorus levels started to show a significant reduction by day 30 since the administration of the tested supplementation in T group: these dogs showed a progressive reduction of serum phosphorus at different time intervals ($p < 0.01$) within the group and compared to the C group. In the present study the tested supplement has proved to be effective, considering that the serum phosphorus in the T group came back to normal (2.7–5 mg/dL) in all dogs within 90 days (12 weeks) or even before, from the beginning of the treatment (Fig. 1).

In our study, the 90% of patients at the time of inclusion in the trial, had a blood bicarbonate concentration below 18 mmol/l and at the end of the study ($T = 180$), the concentration increased significantly ($p < 0.01$) in the T group compared with the C group (Fig. 1, Table 2), but it did not normalize. Although the supplement was able to increase

Martello et al. (2020), *PeerJ*, DOI 10.7717/peerj.9663

**Table 7 Urinary parameters measured during the study.** Urine protein-to-creatinine (UPC), urinary specific gravity(SG), urine protein (UP mg/dL).

| Parameter | Laboratory standard reference range | T0 | T30 | T60 | T90 | T120 | T150 | T180 |
|---|---|---|---|---|---|---|---|---|
| | | | | Parameters group C | | | | |
| UPC | <0.5 | 0.70 ± 0.09 | 0.73 ± 0.04 | 0.75 ± 0.07 | 0.84 ± 0.06 | 0.68 ± 0.04 | 0.74 ± 0.06 | 0.80 ± 0.08 |
| UP mg/dL | 0–150 | 223.57 ± 23.33 | 264.17 ± 28.15 | 272.14 ± 31.75 | 253.33 ± 29.56 | 239.38 ± 24.55 | 249.44 ± 23.18 | 266.88 ± 25.23 |
| SG | 1,020–1,040 | 1,016.43 ± 1.59 | 1,014.17 ± 2.22 | 1,012.14 ± 1.31 | 1,015.56 ± 0.56 | 1015.63 ± 1.39 | 1014.44 ± 1.55 | 1013.13 ± 0.86 |
| | | | | Parameters group T | | | | |
| UPC | <0.5 | 0.63 ± 0.06 | 0.61 ± 0.04 | 0.65 ± 0.04 | 0.67 ± 0.04 | 0.67 ± 0.03 | 0.67 ± 0.03 | 0.65 ± 0.03 |
| UP mg/dL | 0-150 | 314.30 ± 33.91 | 290.10 ± 25.71 | 304.00 ± 23.69 | 302.40 ± 24.02 | 314.80 ± 25.16 | 302.50 ± 22.93 | 301.11 ± 18.09 |
| SG | 1,020–1,040 | 1,014.80 ± 1.74 | 1,016.50 ± 1.30 | 1,014.50 ± 1.39 | 1,015.50 ± 1.39 | 1,016.00 ± 1.00 | 1018.00 ± 1.11 | 1,017.50 ± 1.54 |

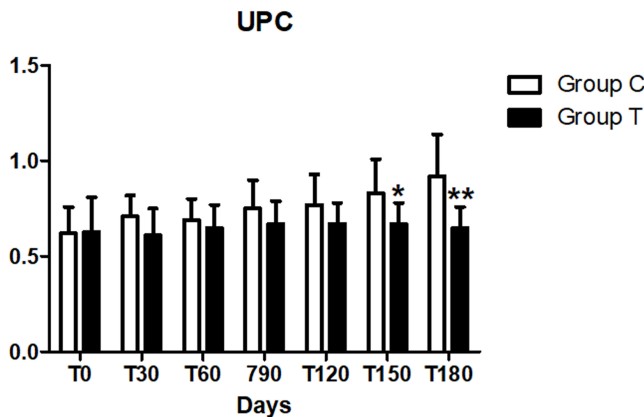

**Figure 4** Urine protein/creatinine (UPC) at days 30, 60, 90, 120, 150 and 180 (*$p < 0.05$, **$p < 0.01$). .

serum concentration of bicarbonate at the recommended dosage, severe conditions of metabolic acidosis may require higher dosages, or extra supplementation of bicarbonate, in order to achieve the goal of a serum bicarbonate concentration > 18 mmol/l.

As CKD dogs usually require both alkalizing and phosphorus binding therapy, the use of a single supplement which includes both agents, like the one we tested, has the advantage to facilitate the owner's compliance. A significant increase of iCa was found in T group at T120, 150 and 180 ($p < 0.01$) compared to the C group (Fig. 1). In literature, hypercalcemia has been reported in animals receiving calcium-based phosphate binders, and it is considered as an adverse event (*Biasibetti et al., 2018*). The phosphate binders used in our supplement were chitosan, calcium lactate gluconate and calcium carbonate. Even if we recorded an increase in iCa, it was not associated with the development of hypercalcemia in the treated dogs. Values remained within the normal range (1.29–1.41 mmol/l) in both dogs with or without a slight hypocalcaemia at the time of the enrolment, indeed we didn't report any adverse clinical sign linked to that condition. At the beginning of the study, only three dogs in the C group were also slightly hypercalcaemic and remained so during the trial.

In general, ionized hypercalcemia and ionized hypocalcemia was reported in dogs and cats with spontaneous CKD (*Polzin, 2011*; *Schenck & Chew, 2003*). Hypercalcemia may occur in dogs with CKD as a consequence of excessive dosages of calcium-containing intestinal phosphate-binding agents (such as calcium acetate, calcium carbonate, or calcium citrate), or in patients with advanced CKD with severe renal hyperparathyroidism (*Finco et al., 1992*; *Nagode, Chew & Podell, 1996*). Hypercalcemia promotes kidney injury and an increase of the calcium X phosphorus product have been correlated to renal mineralization, inflammation, fibrosis and poor prognosis (*Lippi et al., 2014*). Lastly, urinalysis showed a significant change in UPC values in the T group when compared to C group, with significantly decrease at the end of the study, specifically at days 150 and 180 ($p < 0.01$). These findings are different from an analogous study performed on cats testing a similar supplement with calcium carbonate, calcium-lactate gluconate, chitosan and also sodium

bicarbonate (*Biasibetti et al., 2018*), in which UPC showed a significant decrease already at 30 days from the beginning of the study reflecting an overall improvement of the kidney function in cats treated with alkalinisation therapy. The pathogenetic mechanism is controversial, but it may reflect a possible improvement in glomerular filtration rate (GFR).

According to data in human medicine, possible explanations for the reduction of serum concentrations of nitrogen metabolites (BUN and/or CREA), include the compensatory hypertrophy of the functioning nephrons and their enhanced excretion bound of chitosan in the digestive tract. Chitosan can combine acidic substances (i.e., uremic toxins) improving their elimination from the body through faecal material (*Zatelli et al., 2012*).

## CONCLUSIONS

In conclusion, our study design has clearly allowed to see the efficacy of the product on a homogenous class of individuals in IRIS stage 3. The dietary supplementation tested reduced serum phosphorus and increased serum bicarbonate compared to baseline, without causing hypercalcemia, in dogs with CKD IRIS stage 3. It is important to underline that all the animals ($n = 10$) have completed the study and their owners unanimously reported ease of administration of the dietary supplement. Further studies on a larger group of dogs belonging to different IRIS stages would be beneficial.

## ACKNOWLEDGEMENTS

We thank Dr. Mauro Bigliati for reviewing the manuscript.

### Funding
Candioli Pharma S.r.l. provided financial support for this publication. Candioli Pharma S.r.l. is a company that may be affected by the research reported. Candioli Pharma S.r.l. provided the tested supplement, contributed in planning the study and writing the manuscript and it gave financial support for publication.

### Grant Disclosures
The following grant information was disclosed by the authors:
Candioli Pharma S.r.l.

### Competing Interests
Candioli Pharma S.r.l. is a company that may be affected by the research reported. Candioli Pharma S.r.l. provided the tested supplement, contributed in planning the study and writing the manuscript and gave financial support for publication.

### Author Contributions
- Elisa Martello, Francesca Perondi, Ilaria Biasato and Ilaria Lippi analyzed the data, prepared figures and/or tables, authored or reviewed drafts of the paper, and approved the final draft.

- Maria Teresa Capucchio and Natascia Bruni conceived and designed the experiments, authored or reviewed drafts of the paper, and approved the final draft.
- Elena Biasibetti conceived and designed the experiments, analyzed the data, authored or reviewed drafts of the paper, and approved the final draft.
- Tiziana Cocca conceived and designed the experiments, performed the experiments, authored or reviewed drafts of the paper, and approved the final draft.

## Animal Ethics

The following information was supplied relating to ethical approvals (i.e., approving body and any reference numbers):

We followed the Official Italian Guidelines for animal care in scientific studies. In addition, we asked each dog's owner to sign an informed consent to before to participate to the study. The diet supplement used in the study is already on the market. Being the supplement given to our cases a safe product and already on the market, no other declarations are needed.

## Data Availability

Raw data is available as a Supplementary File.

## Supplemental Information

Supplemental information for this article can be found online at http://dx.doi.org/10.7717/peerj.9663#supplemental-information.

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
