# Peer review of "Efficacy of a new dietary supplement in dogs with advanced chronic kidney disease"

_PeerJ, doi:10.7717/peerj.9663_

## Round 0.1 · original submission · Major Revisions

The paper has many flaws in its current state. Kindly check the three reviewers' comments below. The paper would only be accepted after all the concerns and suggestions raised by the reviewers are addressed. Once authors have addressed all the comments the paper will go for another round of review.

Reviewer 1 ·

Basic reporting

- In its current form the are a number of English grammatical errors and sentences that do not read correctly/make complete sense. I would advise getting a native English speaker to proof read the manuscript before any future submissions. Examples include lines 39-40, line 46, line 55, etc.

- The literature references often include review articles, but these are not sufficient to back up specific findings from individual studies. A reader should be able to go straight to the source of the data backing up any points you are making, without having to look up the actual reference in a review article. An example is lines 195-197; the references here should be the actual studies that present this data (ie Elliott 2000, Plantinga 2005, Ross 2006) and not the Polzin review paper.
There are also times when the referencing is not correct, e.g. referencing the de Brito-Ashurst 2009 and Polzin 2013 papers in a sentence about bicarbonate in human medicine (lines 69-71).

- The article has been structured well and includes a number of figures and tables. However, the figures need revising as they have been "cut off" so I can't review the whole figure. The tables also include a very large amount of information, and I don't find it easy to interpret the system of a,b,c indicating statistically significant differences. Please add additional information to explain this.

- The article is self-contained with relevant results to the hypotheses.

Experimental design

- This study does fit within the original primary research Aims and Scope of this journal, although it is not a typical journal for veterinary research,

- The research question is well defined, relevant and meaningful. There is a need for data to support the use of nutritional supplements to address hyperphosphataemia and metabolic acidosis in later stages of CKD.

- The study has been performed rigorously with attention to detail and a large amount of data collected.

- Some additional information needs adding to the methods:
1) Please state the laboratory used for the CBC, biochemistry, urinalysis etc. Please state the machine make used for the venous blood gas analysis. Was ionised calcium not measured on the venous blood gas machine?
2) Please be clear whether dogs were assigned to one specific renal diet, or whether an individual dog could eat any of the renal diets you mention during the study.
3) Does the supplement being tested have a name? If so, please include.
4) Were hypertensive dogs excluded? Please state this. Also, as you were not starting treatment for proteinuria, were dogs with a UPC above a certain number excluded? How do you justify not treating the proteinuria in dogs with a UPC >0.5?

- I would strongly suggest reviewing your statistical methods, because ideally this analysis should have been done using linear mixed models. This would have allowed you to test time vs treatment group more easily, without the need for lots of multiple comparisons and would be easier for you to summarise the data (in figures, rather than huge tables of lots of numbers). If you are not prepared to do this, then we need more information to make it clear where you have applied your Bonferroni correction (and other multiple comparison corrections) - usually, with Bonferroni, the new P value for significance is stated. So where you 7 time points, the P value of 0.05 becomes 0.008: I can't see that you have applied this because you say your superscript letters indicate significant differences at <0.05 between timepoints, but shouldn't these only indicate differences at <0.008?

Validity of the findings

- You have supplied lots of data, but it is difficult to confirm that the data is statistically sound at present (see above).

- Your conclusions are fairly sound, although I am not convinced that changes noted could be due to an improvement in GFR: creatinine that changes <25% is essentially stable given the biological variability of its measurement.

- Please state clearly which IRIS stage the dogs in the study were in. Were they all stage 3, or were some stage 4? If they are all stage 3, then you can say throughout the paper that the supplement was assessed in dogs in IRIS stage 3 (rather than saying "in dogs in advanced stages").

Additional comments

Well done for performing a prospective, randomised, controlled study. Did you consider having a placebo for the control group? You have provided a large amount of data and generally the paper is logically ordered. However, there are a number of points that I would advise you consider (as detailed in the other 3 sections).

Reviewer 2 ·

Basic reporting

This prospective, blindly randomized work (treatment group vs. placebo) aims to describe the efficacy of a feeding stuff in treating metabolic acidosis and hyperphosphatemia in dogs with advanced chronic kidney disease.


It is this reviewer’s opinion the paper reveals numerous lacks concerning either the study setting and compliance with currently ethical standards. Not treating dogs with metabolic acidosis may lead to a sudden worsening of their clinical condition, with complications often leading to death. Contrary to what the authors report, clinical trials of this type require, as long as this type of request is accepted, a specific authorization of an Ethical Committee, a fundamental and missing element in this study. Moreover, the authors do not actually demonstrate a real efficacy of the treatment, rather preferring to repeatedly refer to statistical evaluations between the treatment group and the control one, almost forgetting that the therapeutic purposes are represented by the achievement of endpoints capable of modifying the patient's prognosis. Might the authors state that raising blood bicarbonates from 16.3 to 17.4 mmol / l can actually change the patient's clinical picture and prognosis?
According to the IRIS treatment recommendations, the aim of the treatment should be to maintain serum bicarbonates between 18 and 24 mmol / l, a condition that has not even been achieved in the T group. This reviewer is also curious to know the reasons why the authors decided to treat patients with a dosage of alkalizing of approximately 50% of what suggested in literature. In fact, the product used contains 5 g / 100g of sodium bicarbonate and it was administered at the rate of 0.2g / kg bw orally divided into two daily administrations; assuming the sodium bicarbonate used was 100% pure, each patient was treated with a dose equal to 10mg / kg bw orally divided into two daily administrations (5 mg/kg twice a day) compared to approximately 8-12mg / kg every 8-12 hours which are usually suggested

Experimental design

SEE COMMENTS FOR THE AUTHORS

Validity of the findings

SEE COMMENTS FOR THE AUTHORS

Additional comments

Revision
Line 79-81 When diet alone is insufficient, administration of alkalinizing agents such as potassium citrate, aluminum hydroxide, aluminium carbonate, calcium carbonate, calcium acetate and lanthanum carbonate is indicated…
• The list of raw materials doesn’t include alkalizing agents (eg. “aluminum hydroxide, aluminium carbonate, …. and lanthanum carbonate”). It’s advisable the authors amend the list, reporting just the alkalizing agents.

Line 90 A case-control study on dogs affected by CKD was performed in 2015.,
• Authors should better clarify which criteria were used in classifying patients as affected from CKD;
• Study was performed in 2015, then did the Authors use the IRIS staging system available at that time or the current one (2019)?
• “2015.,” cancel the fullstop “.”

Line 92-94 Dogs affected by other concomitant diseases (acute kidney injury, pre-renal or post-renal azotemia, genitourinary tract inflammation or infection, urinary tract obstruction, chronic heart failure, neoplasia, hypothyroidism, diabetes) were excluded from the study.
• How have the other diseases been diagnosed (laboratory evaluations, instrumental ones, both)?
• Were patients suspected of being affected from other disease excluded too?

Line 96 They provided a written informed consent before the enrollment of their animal.
• Was the study approved from an Ethical Committee which also endorsed the informed consent?

Line 97-98 …dogs were randomly assigned to two groups: the control group (C group, n=10) and the treated group (T group, n=10).
• Which software or system of randomization was used?

Line 103 indirect Doppler
Line 104 method using the radial pulse.
• Join the sentences on the same line

109 …Canine k/d, Hill’s Pet Nutrition)”.” from…
• Cancel “.”


Line 119 Blood Sampling and Assay.
• CBC: neither sampling technique nor the instrument used to process the blood samples undergoing CBC are reported.

Line 120-121 Venous blood gas analysis was also performed in all cases with a standard analytical device.
• It is this reviewer’s opinion there is no analytical device defined as “standard“ to perform a blood gas analysis. As well as they did for the determination of other parameters, Authors should report the device used.

Line 122-126 The laboratory performed CBC and serum biochemical analysis recording the values of: ionized calcium (iCa), serum bicarbonate (HCO3), blood urea nitrogen (BUN), creatinine (CREA), phosphorus (P), total protein (TP), albumin (ALB), albumin/globulin (A/G), glucose (GLU), alanine aminotransferase (ALT), aspartate aminotransferase (AST), alkaline phosphatase (ALP), bilirubin (BIL), and cholesterol (CHOL).
• This reviewer is a bit puzzled. Were either iCa and HCO3 determined through another method than hemogasanalysis? Determining iCa without a blood gas analysis, requires strictly anaerobic methods and if these had been used they should then be described, together with a list of both the devices and methods used. Even the determination of HCO3, if not done with blood gas analysis, should be better described, also considering in the discussion section all the factors potentially influencing this determination. Also determination of HCO3, if not performed through hemogasanalysis, should be better described, considering all the factors potentially influencing this determination in the discussion section.

Line 156-158 Values of haematocrit (HCT), haemoglobin (HG), red blood cells (RBC), white blood cells
(WBC), neutrophil (N), eosinophil (EO), lymphocytes (LYM) (Table 3) were recorded during the entire study.
• These data should be included in M&M rather than in Results. Also timing of sampling and evaluation of samples should be described.

Line 171 No difference in the serum iCa level between groups was reported at day 0.
• Do the Author mean “no statistically significant difference”?

Line 180-182 At the baseline) all dogs (n=10) of the C group, were proteinuric (UPC, 0.5), while 8 dogs in the T group were proteinuric (UPC, 0.5) and 2 were borderline proteinuric (UPC, 0.2 and 0.5) (Table 7).
• Was a statistical evaluation at T0 performed in order to evaluate the difference between groups? If this kind of assessment was carried out, authors should clarify it and report it in the paper; if that was not done, this is a required data.

Line 192 …observation was longer (180 vs 28 days) than other similar studies (Zatelli et al. 2012).
• This reviewer cannot understand this statement. The study of Zatelli et al. 2012, which authors refer to, had a much longer duration (44 weeks) than the study presented (180 days). What do the “28 days” refer to?

Line 201-202 In this study, the values were recorded at 4–8 weeks following the supplement
administration (Zatelli et al. 2012).
• As indicated above, the paper the authors refer to was carried out during a 44 wks period; it needs the concept to be better explained.

Line 212 – 214 The current IRIS guidelines ((IRIS) 2019) for phosphorus control suggests to start phosphate binding therapy if serum phosphorus remains above 5 mg/dL after the initiation of dietary restriction.
• Recommendation above refers to patients in IRIS stage 3. Authors should better explain this aspect.

Line 235-237 …of inclusion in the trial, had a blood bicarbonate concentration below 18 mmol/l and at the end of the study (T=180), blood bicarbonate concentration increased significantly (p<0.01) in the Tgroup compared with the C group (Figure 1, Table 2).
• As it is correctly reported by the Authors, in the case of a patient with metabolic acidosis, the therapeutic aim is to raise bicarbonates above 18 mmol / L and not just to increase them still remaining in an acidotic range (we have no idea if such a result can modify the patient’s prognosis). In order to correctly evaluate the usefulness of the therapy administered, it is necessary to insert a table with values of the single patients as well as offer a statistical determination of the intra-group significance before and after treatment.

Line 247-248 UPC had a significantly decrease at days 150 and 180 in the T group compared with C (p<0.01).
• Based on tables and data provided, it would seem more correct to state that in group C there was an increase in proteinuria, rather than saying in group T this decreased during treatment. The increase of proteinuria in group C could be justified by the higher level of intra-group proteinuria at the time of inclusion compared to group T. A statistical evaluation of the difference between groups at T0 as well as an intra-group one is missing.

Line 252- The pathogenethic mechanism is controversial, but anyway may reflect a possible improvement in glomerular filtration rate (GFR), as also confirmed by the significant reduction in CREA values in the T group compared to the C group (Table 5).
• It is hardly understandable to this reviewer the link between improved GFR and reduction of proteinuria, considering that the product administered is based exclusively on alkalizing (sodium bicarbonate) and intestinal phosphorus binders. It is the opinion of this reviewer that the Authors should better express what is reported.

Line 275-277 All procedures, treatments and animal care were in compliance with the guidelines of the Italian Minister of Health for the care and use of animals (D.L. 4 March 2014 n. 26 and D.L. 27 January 1992 n.116) and UE (Directive 86/609/CEE).
• A clinical trial involving randomization and division into two groups of dogs with advanced chronic kidney disease and exclude
• ing the control group from therapy of metabolic acidosis and hyperphosphataemia (both causes often leading to clinical worsening and death), cannot be carried out without the ethical committee approval.

Reviewer 3 ·

Basic reporting

English language revision would be beneficial, considering the amount of grammatical errors. This does not only make the reading less smooth, but also alters the information that is providing due to the confusing message that is often given.

The structure of the discussion needs to be altered. It should only comment on own results (providing explanations for them and commenting on their relevance, and comparing them to what was already known in literature on that specific matter) and not provide a literature overview again that belongs in the introduction section.

Experimental design

The research is well defined, relevant and meaningful. It should however be more clear which substances of this dietary supplement have been tested before and which have not, so why this is a 'new' dietary supplement. Also, it should be made clear what the goal of each substance is (phosphate binder and/or alkalinizing agent and/or other effect?), this is now not specified anywhere.

It is also not clear whether this is a double-blinded placebo-controlled study, and if so, why this is not mentioned in the title. In the materials and methods section, the authors claim the following: 'Both investigator and owner were not aware of the group assignment'. However, the word 'placebo' is not mentioned anywhere in the article, which is confusing.

Possibly due to language issues, it is often not clear which data have been compared to which date (e.g. are T180-results compared with T0-results within one treatment group, or are the results of the treatment group compared with those of the control group at a certain time point). This is often confusing and hinders clear interpretation of the results.

Lastly, it is not clear whether ethical approval has really been given for this study (see my comment on this in the manuscript)!

Validity of the findings

The discussion is now more a review of literature than a critical approach to the own results, so this part will need adjustments.

Where the author speculates, this was mentioned in the comments in the revision.

Additional comments

Potentially valuable article (especially considering the results 'phosphate binding effect' and 'palatability'), but major revisions are necessary to make it acceptable for publication.

I have not uploaded all tables again, only the manuscript, but it would be preferable to write all laboratory values in SI units.

Annotated reviews are not available for download in order to protect the identity of reviewers who chose to remain anonymous.

---

## Round 0.2 · accepted · Accept

Thanks for addressing all the major conserns of reviewers.

The previous major concern raised by reviewers were on the issue of ethics approval and statistical approach - the authors addressed both issues by explaining in detail. I trust the authors on ethics approval and I am also satisfied with their explanation and changes on the statistical approach.